# All-cause and cause-specific mortality during and following incarceration in Brazil: A retrospective cohort study

Yiran E. Liu[1,2], Everton Ferreira Lemos[3], Christinne Cavalheiro
Maymone Gonçalves[3], Roberto Dias de Oliveira[4], Andrea da Silva Santos[5], Agne
Oliveira do Prado Morais[3], Mariana Garcia Croda[3], Maria de Lourdes Delgado Alves[6],
Julio Croda[3,7,8], Katharine S. Walter[1☯], Jason R. Andrews[1☯]*

1 Department of Medicine, Division of Infectious Diseases and Geographic Medicine, Stanford, California,
United States of America, 2 Cancer Biology Graduate Program, Stanford University School of Medicine,
Stanford, California, United States of America, 3 School of Medicine, Federal University of Mato Grosso do
Sul, Campo Grande, Mato Grosso do Sul, Brazil, 4 State University of Mato Grosso do Sul, Dourados, Mato
Grosso do Sul, Brazil, 5 Faculty of Health Sciences, Federal University of Grande Dourados, Dourados, Mato
Grosso do Sul, Brazil, 6 Division of Prison Health Assistance, Agência Estadual de Administração do
Sistema Penitenciário, Campo Grande, Mato Grosso do Sul, Brazil, 7 Oswaldo Cruz Foundation, Campo
Grande, Mato Grosso do Sul, Brazil, 8 Department of Epidemiology of Microbial Diseases, Yale School of
Public Health, New Haven, Connecticut, United States of America

☯ These authors contributed equally to this work.
* jandr@stanford.edu

## Abstract

### Background

Mortality during and after incarceration is poorly understood in low- and middle-income
countries (LMICs). The need to address this knowledge gap is especially urgent in South
America, which has the fastest growing prison population in the world. In Brazil, insufficient
data have precluded our understanding of all-cause and cause-specific mortality during and
after incarceration.

### Methods and findings

We linked incarceration and mortality databases for the Brazilian state of Mato Grosso do
Sul to obtain a retrospective cohort of 114,751 individuals with recent incarceration.
Between January 1, 2009 and December 31, 2018, we identified 3,127 deaths of individuals
with recent incarceration (705 in detention and 2,422 following release). We analyzed age-
standardized, all-cause, and cause-specific mortality rates among individuals detained in
different facility types and following release, compared to non-incarcerated residents. We
additionally modeled mortality rates over time during and after incarceration for all causes of
death, violence, or suicide. Deaths in custody were 2.2 times the number reported by the
national prison administration ($n = 317$). Incarcerated men and boys experienced elevated
mortality, compared with the non-incarcerated population, due to increased risk of death
from violence, suicide, and communicable diseases, with the highest standardized inci-
dence rate ratio (IRR) in semi-open prisons (2.4; 95% confidence interval [CI]: 2.0 to 2.8),

pmed.1003789

School, UNITED STATES

**Data Availability Statement:** This study utilized
3rd party data provided by the Government of
Brazil, and we are not authorized to share it directly.

The mortality database is provided upon request by Secretária de Vigilância em Saúde (SVS, https://antigo.saude.gov.br/secretaria-svs). The incarceration database is provided by Agência Estadual de Administração do Sistema Penitenciário (AGEPEN, https://www.agepen.ms.gov.br/).

**Funding:** This work was funded by the National Institutes of Health (R01 AI130058 to JRA). The funders had no role in study design, data collection and analysis, decision to publish, or preparation of the manuscript.

**Competing interests:** The authors have declared that no competing interests exist.

**Abbreviations:** AGEPEN, Agência Estadual de Administração do Sistema Penitenciário; CI, confidence interval; DEPEN, Departamento Penitenciário Nacional; IBGE, Brazilian Institute of Geography and Statistics; ICD-10, International Classification of Diseases-10th Revision; IRR, incidence rate ratio; LMIC, low- and middle-income country; RGI, Registro Geral do Interno; SIGO, Sistema Integrado de Gestão Operacional; SIM, Sistema de Informações Sobre Mortalidade; STROBE, Strengthening the Reporting of Observational Studies in Epidemiology.

police stations (3.1; 95% CI: 2.5 to 3.9), and youth detention (8.1; 95% CI: 5.9 to 10.8). Incarcerated women experienced increased mortality from suicide (IRR = 6.0, 95% CI: 1.2 to 17.7) and communicable diseases (IRR = 2.5, 95% CI: 1.1 to 5.0). Following release from prison, mortality was markedly elevated for men (IRR = 3.0; 95% CI: 2.8 to 3.1) and women (IRR = 2.4; 95% CI: 2.1 to 2.9). The risk of violent death and suicide was highest immediately post-release and declined over time; however, all-cause mortality remained elevated 8 years post-release. The limitations of this study include inability to establish causality, uncertain reliability of data during incarceration, and underestimation of mortality rates due to imperfect database linkage.

## Conclusions

Incarcerated individuals in Brazil experienced increased mortality from violence, suicide, and communicable diseases. Mortality was heightened following release for all leading causes of death, with particularly high risk of early violent death and elevated all-cause mortality up to 8 years post-release. These disparities may have been underrecognized in Brazil due to underreporting and insufficient data.

## Author summary

### Why was this study done?

- The global prison population is growing rapidly, particularly in Latin America.

- Incarcerated and formerly incarcerated individuals may be at high risk of death from illness and external causes (i.e., homicide and suicide).

- Mortality during and after incarceration has been described in high-income countries but is poorly understood in low- and middle-income countries (LMICs), including those in Latin America.

- In Brazil, deaths during incarceration may be underreported, and there are no data on deaths after incarceration.

### What did the researchers do and find?

- We retrospectively followed over 114,000 individuals with recent incarceration in a Brazilian state and used database linkage to identify deaths between 2009 and 2018 that occurred during or after incarceration.

- We calculated rates of death from all causes and specific causes among incarcerated and formerly incarcerated individuals, and we compared them to rates of death in the general population.

- We found elevated mortality from violence, suicide, and communicable diseases during incarceration, which was underreported by national and state sources.

- Following release, individuals were at increased risk of mortality from nearly all leading causes of death, with particularly high risk of violent death and suicide among men in the first 2 years post-release.

### What do these findings mean?

- While we could not establish causality between incarceration and mortality, many of our findings indicate increased mortality risk associated with exposure to the carceral environment.

- Accurate and comprehensive data on mortality during and after incarceration are needed in Brazil and other LMICs where this issue remains poorly understood.

- Structural interventions are needed to reduce the mortality risk in this population, including immediate improvement of conditions and healthcare within carceral facilities, supportive programs and services for the early transition period post-release, and implementation of alternatives to incarceration.

## Introduction

The global prison population has increased dramatically over the past 2 decades, with over 11 million people incarcerated in 2018 [1] and an estimated 30 million people worldwide who pass through the carceral system each year [2]. High rates of communicable diseases, chronic medical conditions, mental illnesses, and violence have been documented among incarcerated and formerly incarcerated individuals, attributable to preexisting vulnerabilities as well as to the prison environment and experience of incarceration [3,4]. However, our understanding of the mortality burden resulting from this high morbidity remains incomplete.

Studies on mortality during incarceration in the United States, Europe, and Australia have yielded conflicting results, with some studies showing reduced mortality in custody [5,6], and others reporting elevated mortality [7,8] or differences among subgroups [9,10]. In contrast, there is consensus surrounding the high risk of mortality upon release [11,12], both in the immediate [13–15] and longer term [8,10,16]. Despite this growing body of work, little is known about incarceration and mortality in low- and middle-income countries (LMICs) [17], where the majority of incarcerated and formerly incarcerated populations reside.

The need to understand mortality among currently and formerly incarcerated individuals is especially urgent in South America, which has the fastest growing prison population in the world [1]. In Brazil, the prison population has grown by over 800% since 1990 and is now the third largest in the world [18], with over 740,000 people incarcerated in a prison system built for approximately 460,000 [19]. While most are confined in a "closed" prison system, some individuals with a shorter sentence or who have already served part of their sentence (18% of the prison population) are detained in lower security "semi-open" prisons, where they perform agricultural or industrial work during the day [20]. Approximately 5% of the incarcerated population are detained in police stations [18]; while these facilities are designed for short-term detention of at most a few days, some have reported detentions of months or years in these

settings [20, 21]. Finally, as of 2018, there were over 22,000 children confined in youth detention facilities [22].

Previous studies have reported a high incidence of mental illnesses, communicable diseases, and violence in Brazilian carceral facilities [23–26]. However, the burden of mortality during and after incarceration is poorly understood. Although Brazil's National Prison Department (Departamento Penitenciário Nacional, DEPEN) publishes biannual numbers of deaths in custody, the data are not disaggregated by age or carceral setting, precluding comparative analysis of mortality rates with the general population or across different facility types. Moreover, there is a paucity of data on mortality among formerly incarcerated individuals in Brazil.

Here, we examined deaths in custody and following release in Mato Grosso do Sul, a state in Central Western Brazil with one of the highest incarceration rates in the country (642 per 100,000 residents in 2019) [27]. We linked the state's mortality registry and incarceration database to identify those who were currently or previously incarcerated at the time of death. We compared causes and rates of death across different types of carceral facilities and following release from incarceration, relative to state residents without recent incarceration history.

## Methods

### Study design and data sources

We carried out a retrospective cohort study on mortality during and following incarceration in Mato Grosso do Sul state by linking the mortality registry with the state's incarceration database. We obtained permission to access the Brazilian mortality registry, Sistema de Informações Sobre Mortalidade (SIM), for Mato Grosso do Sul state. SIM contains individual-level demographic and cause of death information for all notified deaths but lacks information on incarceration status and history. We additionally obtained permission from the Mato Grosso do Sul state prison administration, Agência Estadual de Administração do Sistema Penitenciário (AGEPEN), to access Sistema Integrado de Gestão Operacional (SIGO) from January 1, 2005 to December 31, 2018. Maintained by police in Mato Grosso do Sul, SIGO is a state database of all individual-level movements (i.e., arrests, court appearances, transfers, releases, and deaths) within the state's criminal legal system. While SIGO documented some deaths of incarcerated individuals between 2005 and 2018, it lacked demographic and cause of death information.

We limited the study period to 2009 to 2018 to account for individuals in the carceral system who did not have a documented movement prior to 2009, and we applied additional inclusion criteria to obtain a cohort of 114,751 individuals with (1) at least 1 incarceration in closed prison, semi-open prison, police station detention, or youth detention between 2009 and 2018; and/or (2) at least 1 interval in the community following release from closed or semi-open prison between 2009 and 2018. We include police station detention as a form of incarceration, as some individuals are detained in police stations for extended periods pretrial and post-sentencing due to severe prison overcrowding [20,21]. For individuals with multiple incarcerations and/or releases, all intervals during the study period were included.

We report this study as per the Strengthening the Reporting of Observational Studies in Epidemiology (STROBE) guidelines (S1 Checklist).

We also retrieved publicly available, aggregate data from the national DEPEN information system [19] on prison deaths by state; in Mato Grosso do Sul, this information is entered by the state prison administration (AGEPEN). We did not have a prespecified analysis plan.

### Ethics statement

This study was approved by Federal University of Mato Grosso do Sul (CAAE: 20531819.5.0000.0021) and the Stanford University Institutional Review Board (IRB-50466).

## Preprocessing of SIGO incarceration database

To identify unique individuals within SIGO, we compared internal identification numbers (Registro Geral do Interno, RGI) and performed fuzzy string matching on names and mother's names using the *fuzzywuzzy* package in Python [28] (S1 Text, S1 Fig). Ultimately, we identified 143,551 unique individuals with at least 1 record in SIGO between 2005 and 2018; these individuals were excluded from the general population to obtain the population without recent incarceration history. Additional criteria specified above were applied to generate the cohort of 114,751 individuals. As all analyses were sex stratified, individual-level sex in SIGO was inferred using facility information and manual evaluation (S1 Text).

Movement data in SIGO were converted to interval data (S1 Text). Intervals of incarceration in each facility type were censored upon transfer to a different facility type, release, death, or the end of the study period. Exceedingly long intervals in police stations, semi-open prisons, and youth detention were truncated to account for underreporting of releases from these facilities (S1 Text). Intervals in the community following release from prison (closed or semi-open) were censored upon reincarceration in any facility type, death, or the end of the study period.

## Linkage of SIGO incarceration database and SIM mortality registry

To link SIGO and SIM, we extracted names and mother's names from each database and performed fuzzy matching (S1 Text). At the optimal similarity threshold, our matching algorithm had a sensitivity of 76.6%, specificity of 99.8%, and positive predictive value of 94.7%. To further minimize false positives, we performed additional filtering and manual corrections (S1 Text, S2 Fig). Of deaths documented in SIGO, 34 (9.1%) had no match in SIM and were included in all-cause instantaneous hazard analyses but excluded from age- and cause-based analyses.

## Determination of causes of death

Causes of death in SIM were classified using the International Classification of Diseases-10th Revision (ICD-10). To enhance comparability with other cause of death analyses, we mapped ICD-10 codes to public health cause of death categories using a previously defined list that builds on methods used by the Global Burden of Disease Study [29], with some exceptions (S1 Text). Of note, 27 deaths of incarcerated men were reported as "hanging, strangulation and suffocation, undetermined intent"; we classified this cause under interpersonal violence. Deaths were considered of unknown cause if there was no record in SIM or if the reported cause of death was "other ill-defined and unspecified causes of mortality" or "unspecified event, undetermined intent." Finally, we assigned intermediate categories to the following broad cause of death categories: violence, suicide, communicable diseases, noncommunicable diseases, or transport injuries (S1 Text).

## Person-time and age structure of the incarcerated population

We summed intervals of incarceration in each facility type, censored as described above, to obtain cumulative person-time of incarceration in each facility type between 2009 and 2018. We estimated age structure as follows. The annual age structure of the incarcerated population in Mato Grosso do Sul was reported by DEPEN using wide age bins. To obtain more granular estimates of age structure, we fit truncated negative binomial distributions to the binned age data using the R packages *fitdistrplus*, *truncdist*, and *revengc* [30–32] (S1 Text), with separate distributions by sex and calendar year. We adjusted the modeled counts to correspond to the reported proportions within each wide age bin. We then applied the adjusted age structure to

obtain cumulative age-stratified person-time for each facility type. We assumed the same age structure in closed prisons, semi-open prisons, and police stations due to lack of facility type-specific age structure. For analyses of risk over incarceration time, we first applied the annual age structure of the incarcerated population, as estimated above, to person-time stratified by years of incarceration, shifting the age structure 1 year older for each elapsed year of incarceration.

## Person-time and age structure of the formerly incarcerated population

We summed post-release intervals, censored as described above, and stratified person-time by calendar year upon release and years elapsed post-release. For example, the cohort of men released from prison in 2012 contributed approximately 6,300, 4,700, and 4,000 person-years in years 0, 1, and 2 post-release, respectively. To estimate age structure, we first applied the estimated annual age structure of the incarcerated population, shifted 1 year older for each year elapsed post-release, to the person-time of the formerly incarcerated population stratified by year of release and years elapsed post-release. To account for changes in the formerly incarcerated population age structure over time due to deaths after incarceration, we subtracted age-specific death counts, also stratified by year of release and years elapsed post-release. We then computed the sum of age-stratified person-time over the study period to obtain cumulative age-stratified person-time for the formerly incarcerated population.

## Person-time and age structure of the non-incarcerated population

For the population without recent incarceration history (hereafter referred to as the "non-incarcerated population"), we used projected binned age structure for Mato Grosso do Sul from the Brazilian Institute of Geography and Statistics (IBGE) [33] and estimated cumulative age-specific person-time using the same method as described for the incarcerated population. We excluded individuals with any record in SIGO from the non-incarcerated population by subtracting person-time of the incarcerated and formerly incarcerated population, the latter of which was recomputed using releases from any facility type (not only releases from prison).

## Statistical analysis for mortality rates

We calculated age-specific rates of all-cause and cause-specific mortality by dividing the number of deaths in each age group by the cumulative age-specific person-years of incarceration in each carceral facility type or post-release. For age-standardized rates, we computed age-specific rates using 4-year age bins and performed direct standardization to the incarcerated population. We excluded deaths of individuals under 18 years of age in closed prisons, semi-open prisons, police stations, and post-release due to missing age structure information for those under 18. However, we were unable to exclude individuals under 18 from the study cohort due to lack of individual-level age information in SIGO. All deaths in youth detention were included (ages 14 to 19), from which we calculated crude rates in the absence of age structure information. To compare rates, we computed age-specific and age-standardized incidence rate ratios (IRRs) relative to the non-incarcerated population. Exact confidence intervals (CIs) for age-specific rate ratios were computed using the 2-sample Poisson exact test. Approximate CIs were computed for age-standardized rate ratios using the F distribution [34].

## Survival analyses

To model instantaneous mortality rates over time, we used the *bshazard* package in R to perform nonparametric estimation of the hazard function [35]. We estimated separate hazard

functions for deaths resulting from all causes, deaths resulting from violence or suicide, and deaths resulting from other causes.

To analyze the association between total length of incarceration and mortality post-release, we generated a reduced cohort of men whose last recorded release from prison was at least 5 years before the end of the study period ($N = 23,608$). Post-release intervals longer than 5 years were censored at 5 years to obtain consistent periods of observation. We computed each individual's cumulative time incarcerated in any facility type prior to release and estimated survival from violent death or other causes using Kaplan–Meier, with log-rank tests for significance (*survival* package in R [36]). While we were unable to perform age-adjusted survival analysis due to lack of individual-level age information in SIGO, we examined within-age-group associations of cumulative incarceration time with cause of death and time to death post-release.

All analyses are summarized in S1 Table.

## Results

### Linkage of incarceration and mortality databases reveals deaths during detention and following release

Our cohort consisted of 114,751 individuals (105,465 men and 9,286 women) with at least 1 incarceration in any facility type and/or at least 1 interval in the community following release from prison between 2009 and 2018 (Table 1). Together, they contributed 133,879 total person-years of incarceration (103,992 in closed prisons, 17,428 in semi-open prisons, 8,292 in police stations, and 4,167 in youth detention) and 302,420 person-years in the community following release from prison. By linking the incarceration database (SIGO) with the state mortality registry (SIM), which recorded 133,923 deaths of individuals aged 14 to 95, we identified 3,127 (2.3%) deaths, including 705 (0.5%) deaths among people who were currently incarcerated and 2,422 (1.8%) deaths that occurred in the community following release from prison. Among those who died while incarcerated, 442 (62.7%) individuals died in closed prisons, 130 (18.4%) in semi-open prisons, 88 (12.5%) in police stations, and 45 (6.4%) in youth detention facilities (Fig 1, S3 Fig). In total, we identified 2.2 and 1.9 times as many deaths in custody between 2009 and 2018 as reported by the national prison administration DEPEN ($N = 317$ deaths) and SIGO ($N = 375$), respectively (S4 Fig). Notably, DEPEN reports conflicted with SIGO data each year.

### Leading causes of death varied by age, sex, incarceration status, and facility type

Leading causes of death varied by age, sex, and incarceration status and included interpersonal violence, cardiovascular disease, transport injuries, neoplasms, suicide, respiratory infections including tuberculosis, and HIV/AIDS (S2 Table, Fig 2, S5 Fig). Proportions of cause-specific deaths also varied by carceral facility type. For example, suicides constituted a larger proportion of deaths among men in police stations ($N = 15$; 17.6%) than in closed prisons ($N = 34$; 8.1%) or semi-open prisons ($N = 4$; 3.3%) (Fig 2). While the proportion of violent deaths decreased with increasing age across all populations, they nonetheless contributed disproportionately to deaths among incarcerated and formerly incarcerated individuals across age groups compared to non-incarcerated individuals (Fig 2, S5 Fig).

### Mortality rates among incarcerated men in closed prisons

The all-cause mortality rate for men in custody was elevated across nearly all age-groups and carceral facility types compared to non-incarcerated men (S6 Fig). For men detained in closed

**Table 1. Characteristics of study cohort.**

| | | Men and boys (*N* = 105,465) | Women (*N* = 9,286) |
|---|---|---|---|
| **Number of individuals (%) with 1+ intervals in each location** | Closed prisons | 60,285 (57.2) | 7,715 (83.1) |
| | Semi-open prisons | 26,667 (25.3) | 3,233 (34.8) |
| | Police stations | 72,339 (68.6) | 4,332 (46.7) |
| | Youth detention | 4,939 (4.7) | 18 (0.2) |
| | Post-release | 60,184 (57.1) | 8,347 (89.9) |
| **Median duration in days (IQR)** | Closed prisons | 153 (21 to 473) | 184 (34 to 482) |
| | Semi-open prisons | 62 (13 to 169) | 93 (29 to 240) |
| | Police stations | 3 (1 to 17) | 2 (1 to 21) |
| | Youth detention | 229 (27 to 758) | 300 (46 to 498) |
| | Post-release | 348 (50 to 1,300) | 849 (131 to 2,262) |
| **Percentage of incarcerated population in each age group** | 18 to 24 | 23.3 | 26.5 |
| | 25 to 29 | 24.4 | 20.6 |
| | 30 to 34 | 19.8 | 19.0 |
| | 35 to 45 | 22.3 | 23.0 |
| | 46 to 60 | 8.9 | 9.8 |
| | 61+ | 1.3 | 1.1 |
| **Number of deaths (% of total deaths)** | Under 18 | 27 (0.9) | 0 (0) |
| | 18 to 24 | 423 (14.5) | 14 (6.4) |
| | 25 to 29 | 396 (13.6) | 22 (10.0) |
| | 30 to 34 | 389 (13.4) | 23 (10.5) |
| | 35 to 45 | 658 (22.6) | 73 (33.3) |
| | 46 to 60 | 608 (20.9) | 54 (24.7) |
| | 61+ | 374 (12.9) | 31 (14.2) |
| | Unreported | 33 (1.1) | 2 (0.9) |
| | Total | 2,908 | 219 |

Percentages of individuals in each location sum up to >1 as many individuals had intervals in more than 1 location; other percentages may not sum up to 1 due to rounding. Raw duration data are shown, prior to interval truncation. Annual binned age structure data (in proportions) for the prison population in Mato Grosso do Sul were accessed from the national prison administration (DEPEN) and weighted by person-time of incarceration each year to generate the summarized age group percentages shown.

1+, 1 or more; DEPEN, Departamento Penitenciário Nacional; IQR, interquartile range; post-release, post-release from prison (closed or semi-open).

prisons, the all-cause age-standardized mortality rate was 423 (95% CI: 383 to 467) per 100,000 person-years (Fig 3). Relative to non-incarcerated men (321 per 100,000; 95% CI: 317 to 325), the age-standardized IRR of mortality for men in closed prisons was 1.3 (95% CI: 1.2 to 1.5), driven by elevated rates of death from violence (IRR = 2.2; 95% CI: 1.8 to 2.6), suicide (IRR = 2.3; 95% CI: 1.6 to 3.2), and communicable diseases (IRR = 3.0; 95% CI: 2.4 to 3.8) and partly offset by reduced rates of death from transport injuries (IRR = 0.1, 95% CI: 0.05 to 0.2) (Fig 3). Among deaths from communicable diseases, the disparity was greatest for deaths from tuberculosis, with a rate ratio of 7.1 (95% CI: 4.0 to 11.9) relative to non-incarcerated men.

## Mortality rates among incarcerated men and boys in other facility types

Among men incarcerated in semi-open prisons, the all-cause age-standardized mortality rate was 762 (95% CI: 630 to 913) per 100,000 person-years or 2.4 times (95% CI: 2.0 to 2.8) that of non-incarcerated men (S7 Fig). Severely elevated rates of violent deaths (IRR = 7.3, 95% CI: 5.5 to 9.5) contributed most to the disparity in all-cause mortality in semi-open prisons. In police stations, the all-cause age-standardized rate was even higher, at 1,006 deaths (95% CI:

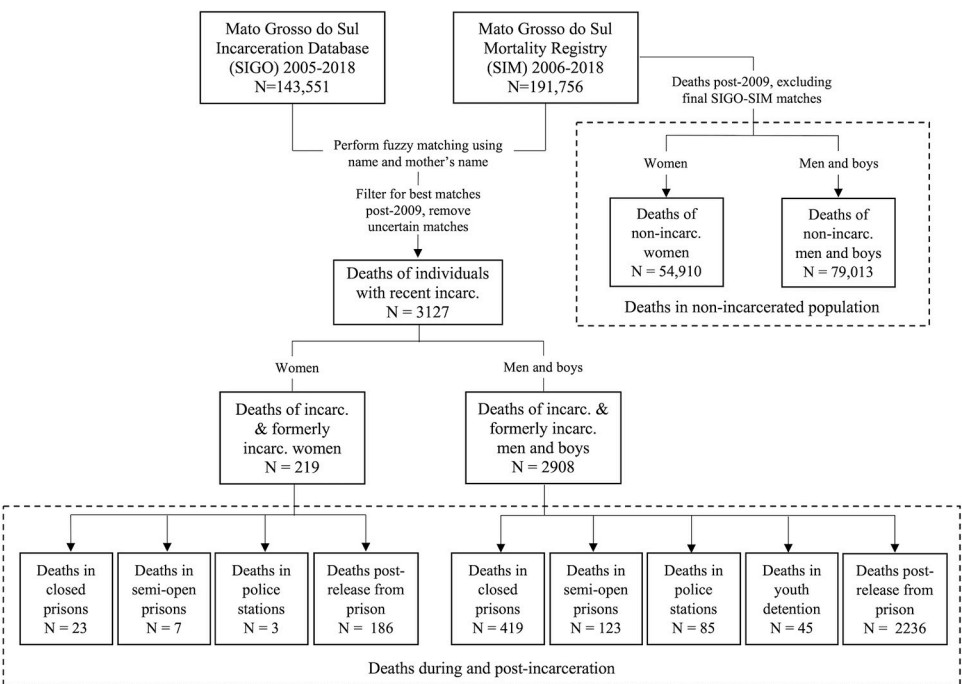

**Fig 1. Linkage of incarceration database and mortality registry to identify deaths during and following incarceration in Mato Grosso do Sul state.** Deaths of men and boys ages 14 to 95 and women ages 18 to 95 were included. See S2 Fig and S1 Text for filtering and inclusion/exclusion details. incarc, incarcerated/incarceration; SIGO, Sistema Integrado de Gestão Operacional (incarceration database); SIM, Sistema de Informações Sobre Mortalidade (mortality registry).

793 to 1258) per 100,000 person-years. This rate was 3.1 (95% CI: 2.5 to 3.9) that of non-incarcerated men; this disparity was driven by exorbitant rates of death from violence (IRR = 7.9, 95% CI: 5.4 to 11.3) and suicide (IRR = 12.4, 95% CI: 6.9 to 20.7). The most extreme disparity in all-cause and violent death rates was among boys in youth detention facilities, at 8.1 (95% CI: 5.9 to 10.8) and 19.4 (95% CI: 13.3 to 27.5) times, respectively, those of non-incarcerated boys. The risk of death declined with incarceration time in police stations but increased with incarceration time in semi-open prisons and youth detention (S8 and S9 Figs).

## Mortality rates among incarcerated women

For incarcerated women, the number of deaths was too small to perform stratified analyses by facility type; we therefore combined deaths across facility types. The all-cause age-standardized mortality rate among incarcerated women in any facility type was 268 (95% CI: 182 to 380) per 100,000 person-years, comparable to non-incarcerated women (291 per 100,000; 95% CI: 285 to 296) (Fig 3). While incarcerated women were less likely to die from noncommunicable diseases (IRR = 0.6, 95% CI: 0.3 to 1), they were at increased risk of death from suicide (IRR = 6.0, 95% CI: 1.2 to 17.7) and communicable diseases (IRR = 2.5, 95% CI: 1.1 to 5.0).

## Mortality rates among formerly incarcerated men and women

Following release from prison, the all-cause mortality rates for men and women were 947 (95% CI: 905 to 992) and 363 (95% CI: 310 to 427) per 100,000, respectively (Fig 4). These rates were 3.0 (95% CI: 2.8 to 3.1) and 2.4 (95% CI: 2.1 to 2.9) times those of men and women with no recent incarceration, respectively, and were driven by elevated rates from nearly all

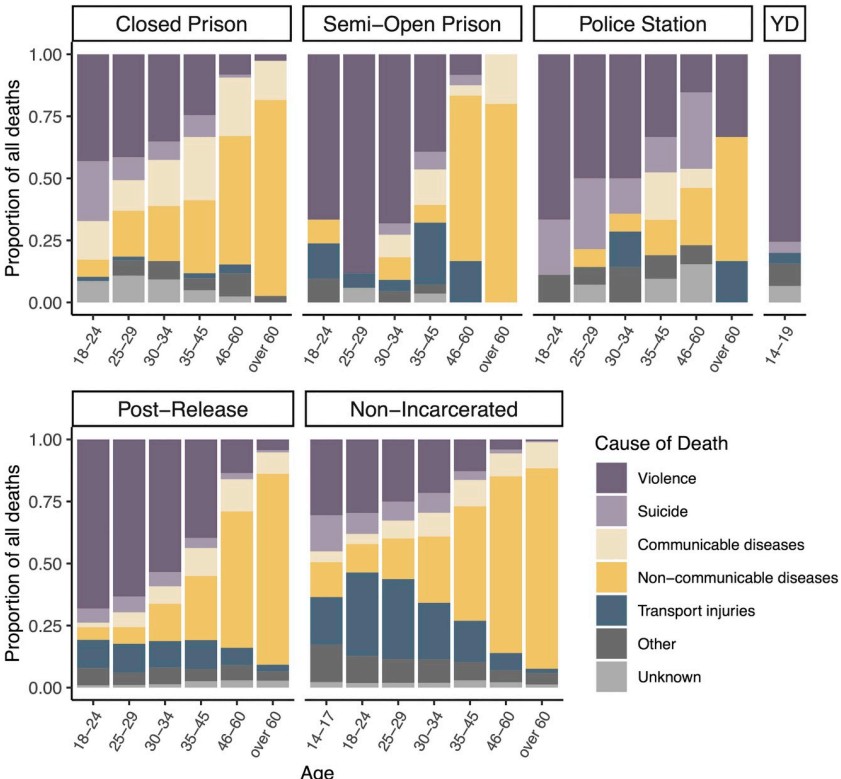

**Fig 2. Proportions of cause-specific deaths by age, carceral facility, and incarceration history.** Stacked bar plots indicating the proportions of deaths of men and boys resulting from each cause category, stratified by age group, carceral facility type, and incarceration status. Post-release refers to release from prison. Non-incarcerated refers to male Mato Grosso do Sul residents who did not have a record in the state's incarceration database between 2005 and 2018. YD, youth detention.

leading causes of death and across all age groups (Fig 4, S6 and S10A Figs). Violent death rates exhibited the greatest disparity between formerly incarcerated individuals and non-incarcerated individuals, with an IRR of 7.2 (95% CI: 6.7 to 7.8) and 9.3 (95% CI: 6.2 to 14.4) for formerly incarcerated men and women, respectively.

The all-cause mortality rate among formerly incarcerated men was highest immediately after release and declined sharply in the first 2 years following release from prison (Fig 5). This effect was largely due to the rapid reduction in rates of violent death and suicide in the first 2 years post-release (S11 Fig). Younger age was associated with greater proportions of violent deaths and shorter time to death post-release (S11 Fig). Furthermore, among men under age 46, increasing cumulative time incarcerated in any facility type was associated, across age groups, with increased odds of early death from violence and other causes following release.

Among formerly incarcerated women, the violent death and suicide rate declined steadily, while mortality from other causes increased (S10B Fig). For both men and women, the all-cause mortality rate remained elevated 8 years after release from prison at approximately twice that of non-incarcerated men and women, respectively (Fig 5, S10B Fig).

## Discussion

Mortality during and after incarceration remains poorly understood in LMICs. In Brazil, the incarcerated population has undergone the largest absolute increase since 2000 of any country

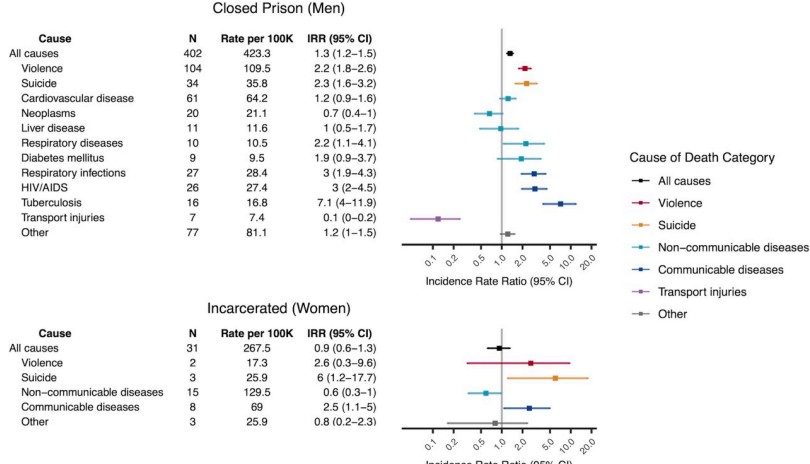

**Fig 3. Age-standardized mortality rates among incarcerated men and women.** All-cause and cause-specific mortality rates and IRRs for the leading causes of death among incarcerated men in closed prisons and for broad causes of death among incarcerated women in any facility type. Rates were directly standardized to the age structure for incarcerated men or women. IRRs were computed relative to non-incarcerated male or female Mato Grosso do Sul residents. Colors correspond to broad cause of death categories. Within each category, causes are ordered by number of deaths. A total of 17 deaths of men in closed prisons and 2 deaths of incarcerated women were excluded due to missing age information. 100K, 100,000 person-years; IRR, incidence rate ratio; *N*, number of deaths.

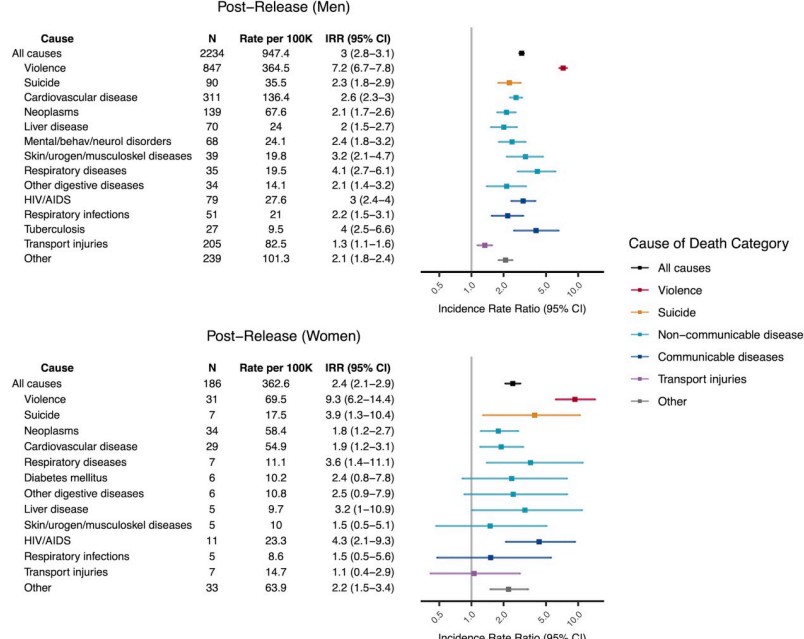

**Fig 4. Age-standardized mortality rates among formerly incarcerated men and women.** All-cause and cause-specific mortality rates and IRRs for the leading causes of death among formerly incarcerated men and women. Rates were directly standardized to the age structure for incarcerated men or women. IRRs were computed relative to non-incarcerated male or female Mato Grosso do Sul residents. Colors correspond to broad cause of death categories. Within each category, causes are ordered by number of deaths. A total of 2 deaths of formerly incarcerated men were excluded due to missing age information. 100K, 100,000 person-years; IRR, incidence rate ratio; *N*, number of deaths; mental/behav/neurol disorders, mental/behavioral, neurological, and sensory organ disorders; skin/urogen/musculoskel diseases, skin/genitourinary/musculoskeletal diseases.

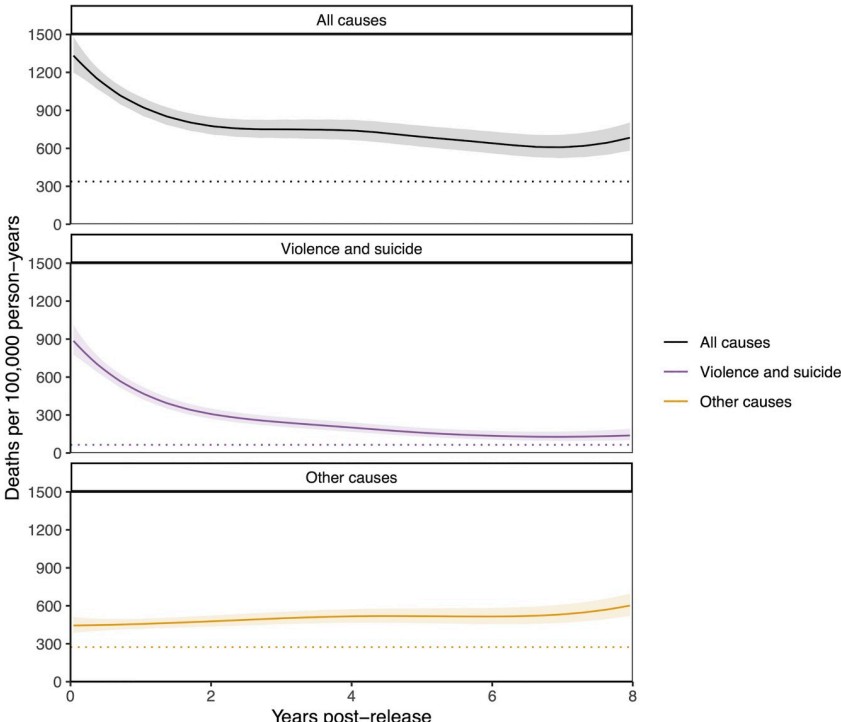

**Fig 5. All-cause and violent death rates over time among men following release from prison.** Instantaneous crude death rate among formerly incarcerated men for all causes of death (black), violence and suicide (purple), or causes other than violence and suicide (yellow). Bands indicate the 95% CI. The dotted horizontal line indicates the age-standardized mortality rate among non-incarcerated male residents of Mato Grosso do Sul, standardized to the age structure of the formerly incarcerated population. CI, confidence interval.

in the world [37]. By linking the incarceration and mortality databases in Mato Grosso do Sul state, we identified approximately twice as many deaths in custody between 2009 and 2018 as reported by national and state sources. We found that incarcerated men and boys had markedly higher all-cause mortality compared with non-incarcerated men and boys, due to increased risk of death from violence, suicide, and communicable diseases. Incarcerated women exhibited reduced mortality from noncommunicable diseases but higher rates of suicide and communicable disease deaths than non-incarcerated women. Among formerly incarcerated men and women, mortality rates were raised for nearly all leading causes of death, with particularly high risk of violent death early post-release among younger men with longer total incarceration time. Strikingly, all-cause mortality remained considerably elevated for men and women 8 years post-release. Together, these results demonstrate the substantial mortality burden among incarcerated and formerly incarcerated individuals in Brazil that may have been previously underrecognized due to insufficient data.

## Mortality during incarceration

Incarcerated men experienced heightened risk of all-cause mortality across all facility types and nearly all age groups, driven by elevated rates of death from violence, suicide, and communicable diseases. While these findings were consistent with some studies from high-income countries that reported increased risk of homicide [8,38], suicide [39–41], and infectious disease deaths [6,10] during incarceration, they contrast with other studies that reported reduced

all-cause mortality and homicide during incarceration [5], particularly among subgroups with higher baseline mortality [9,10,42,43]. The root causes of these differences are undoubtedly complex and were not addressed by the present study but may include varying prison conditions, management, and culture.

We additionally found that causes and rates of death varied considerably among men and boys in different carceral facility types, reflecting different environments and populations in each setting. Communicable disease mortality was most elevated in closed prisons, where circumstances may especially facilitate transmission. Suicide rates were highest among men in police stations, where extreme conditions and torture have been described [20]. However, given potential underreporting of violent deaths in Brazil [44], it is possible that some violent deaths were incorrectly classified as suicide, particularly in settings where administrative oversight may be limited. Finally, boys in youth detention had exceptionally high violent death rates compared to non-incarcerated boys, a disturbing finding in light of recent proposals to expand youth detention in Brazil [45,46].

Although women make up a smaller portion of the incarcerated population, the population of incarcerated women in Brazil has increased by 368% over the past 2 decades [37]. Incarcerated women had a greater proportion of disease-related deaths compared to incarcerated men, with reduced mortality from noncommunicable diseases and elevated mortality from communicable diseases and suicide relative to non-incarcerated women. While we were limited by the relatively small number of deaths among incarcerated women, future work might explore sex-specific effects of incarceration on mortality in Brazil [47].

Finally, we found that deaths in custody were underreported by and inconsistent between national (DEPEN) and state (SIGO) sources. These discrepancies were not explained by the lack of DEPEN data on deaths in police stations or youth detention but may instead indicate differences in mission, operations, and management between the agencies responsible for data reporting (state police for SIGO, state prison administration for DEPEN).

## Mortality post-release

Following release, all-cause mortality was elevated among men and women of all ages for nearly all leading causes of death, with particularly high violent death and suicide rates in the early period that declined sharply in the first 2 years post-release. Among men, younger age and increasing cumulative incarceration time were associated with early post-release death, especially from violence. Our findings mirror trends in high-income countries [13,16,48], where younger age, male sex, length of incarceration, and/or repeated incarceration were also risk factors for early post-release mortality [8, 13]; however, in contrast to studies in high-income countries [11,15], drug overdose only comprised a small fraction of post-release deaths in our cohort. Therefore, additional research is needed to identify risk factors for post-release mortality in settings with different social conditions and leading causes of death.

Upon release, formerly incarcerated men and women also experienced elevated disease mortality that further increased over time. This disparity was a result of increased communicable and noncommunicable disease mortality post-release and corresponds to studies reporting high incidence of infectious diseases [49,50] and chronic medical conditions [51] among formerly incarcerated individuals. Recent work in Mato Grosso do Sul found bacterial genomic evidence of new tuberculosis diagnoses among formerly incarcerated individuals that could be traced to transmission within carceral facilities [52], indicating that elevated communicable disease mortality post-release may at least be partially attributable to incarceration. However, more data are needed to determine the respective contribution of incarceration versus preexisting vulnerabilities to disease mortality post-release.

## Implications

The extreme mortality burden associated with incarceration in Brazil necessitates urgent actions. First, we identify a clear need for accurate, comprehensive data on mortality during and after incarceration. Second, our findings of elevated violent mortality during and after incarceration are particularly grave in the context of exorbitant homicide rates in Brazil at large [53]. Importantly, they demonstrate the failure of incarceration to reduce violence, even with purported reforms such as the semi-open regime that was designed to facilitate societal reentry. Instead, alternatives to incarceration are needed that repair interpersonal harms and deep-rooted social problems without endangering the lives of an already vulnerable population. Third, the high risk of disease-specific mortality and suicide during and after incarceration warrants immediate improvement of conditions within Brazilian carceral facilities, including reducing overcrowding and expanding access to primary and mental healthcare. Finally, the heightened risk of mortality immediately following release highlights the need for individual and structural interventions during this vulnerable transition period.

## Limitations

Our study has several limitations. First, in the absence of a counterfactual, we were unable to establish causality between exposure to incarceration and the mortality trends described. Insufficient data also precluded our ability to adjust for socioeconomic factors, pretrial status, conviction type, or other potential confounders. Second, cause of death classifications during incarceration may have reduced reliability, as evidenced by the greater proportion of deaths of unknown or imprecise cause among incarcerated individuals compared with non-incarcerated individuals. Third, our approach likely underestimates mortality rates, due to (1) underreporting of deaths in custody to the mortality registry; (2) incomplete incarceration and age data and uncertain location or status at time of death; (3) migration out of Mato Grosso do Sul following release; and (4) our matching algorithm for database linkage that optimized specificity (99.8%) at the expense of sensitivity (76.6%). A sensitivity analysis including uncertain matches by name did not substantially change our conclusions (S1 Text, S12 Fig). Next, we did not have facility type-specific age structure information; consequently, our assumption of the same age structure across facility types may have affected rate estimates. We were additionally unable to conduct stratified analyses by race/ethnicity due to insufficient data, but this should be a priority area of future research given Brazil's disproportionate incarceration of Black and mixed-race people [19]. Furthermore, many individuals died after release from facility types other than prison; these deaths were excluded from our analyses of mortality post-release from prison. Finally, our findings may not be generalizable to other states or countries. However, Mato Grosso do Sul is likely not an outlier in Brazil: the crude mortality rate during incarceration in Mato Grosso do Sul, as reported by DEPEN, was similar to the mean national rate (S13 Fig).

## Conclusions

In this statewide retrospective cohort study that followed over 114,000 incarcerated and formerly incarcerated individuals over a 10-year period, we found a considerable burden of mortality among individuals in detention and in the community after release, which may have been previously underrecognized due to limited and underreported data. As the first systematic study, to our knowledge, of mortality during and after incarceration in Latin America and in LMICs more broadly, our findings both resemble and differ from previously reported trends in high-income countries, highlighting the need for improved documentation of incarceration-associated mortality in LMICs. Finally, our study underscores the urgent need for

structural reforms, social programs, and public health interventions that protect the health and lives of this vulnerable population.

## Supporting information

**S1 Checklist. STROBE guideline checklist.** STROBE, Strengthening the Reporting of Observational Studies in Epidemiology.
(DOCX)

**S1 Text. Details on data preprocessing, database linkage, cause of death classification, age structure estimation, and sensitivity analysis.**
(DOCX)

**S1 Table. Overview of analyses by population and facility type/location.** List of analyses and figures, organized by incarceration status (during incarceration or following release), group (men, boys, or women), and facility type or location. Reference population used for computing IRRs is shown. IRR, incidence rate ratio.
(PDF)

**S2 Table. Leading causes of death, by incarceration status and sex.** Causes are sorted top to bottom by total number of deaths among all groups. Proportions of deaths resulting from each cause within each group (columns) are indicated in parentheses. Top 5 causes of death within each group are indicated by a number sign (#). Percentages may not sum to 100% due to rounding. TB, tuberculosis.
(PDF)

**S1 Fig. Decision trees for identifying unique individuals within the SIGO incarceration database.** Logic for determining whether a set of SIGO records correspond to the same unique individual in the case of records with **(A)** the same RGI number (RGI, an internal identification number) but multiple names or **(B)** the same name but multiple RGI numbers. String similarity between names and mother's names was computed using Levenshtein distance, weighted based on difference in length between strings. RGI, Registro Geral do Interno; SIGO, Sistema Integrado de Gestão Operacional.
(PDF)

**S2 Fig. Overview of selection and filtering steps for matches between the SIGO incarceration database and SIM mortality registry.** Arrows to the far right indicate exclusion. SIGO, Sistema Integrado de Gestão Operacional; SIM, Sistema de Informações Sobre Mortalidade.
(PDF)

**S3 Fig. Deaths in custody and person-years of incarceration in each facility type throughout the study period.** Bar plots depicting number of deaths of individuals during incarceration in each facility type (left axis), overlaid with line plots of person-years of incarceration in each facility type (right axis).
(PDF)

**S4 Fig. Numbers of deaths in custody each year as reported by national and state sources or identified by database linkage.** Bar plot of deaths during incarceration each year identified through database linkage in this study (yellow), reported by the Brazilian National Prison Department (DEPEN; light blue), or documented in Mato Grosso Do Sul's incarceration database (SIGO; dark blue). DEPEN, Departamento Penitenciário Nacional; SIGO, Sistema Integrado de Gestão Operacional.
(PDF)

**S5 Fig. Proportions of cause-specific deaths among women vary by incarceration status.**
Stacked bar plots indicating the proportions of deaths of women resulting from each cause category, stratified by age group and incarceration status. General population refers to female Mato Grosso do Sul residents who did not have a record in the state's incarceration database between 2005 and 2018. Incarc, incarcerated.
(PDF)

**S6 Fig. All-cause mortality rates for incarcerated and formerly incarcerated men by age.**
Age-specific all-cause mortality rates per 100,000 person-years and IRRs for incarcerated and formerly incarcerated men. IRRs were computed relative to non-incarcerated male Mato Grosso do Sul residents. 100K, 100,000 person-years; IRR, incidence rate ratio; *N*, number of deaths.
(PDF)

**S7 Fig. Mortality rates among men and boys in other carceral facility types.** All-cause and cause-specific mortality rates per 100,000 person-years and IRRs for men and boys detained in semi-open prisons, police lockups, and youth detention. Rates were directly standardized to the age structure among incarcerated men. Crude, age-specific rates are depicted for boys 14 to 19 years of age in youth detention. IRRs were computed relative to non-incarcerated male Mato Grosso do Sul residents. A total of 6 deaths in semi-open prisons and 8 deaths in police stations were excluded due to missing age information. 100K, 100,000 person-years; IRR, incidence rate ratio; *N*, number of deaths.
(PDF)

**S8 Fig. All-cause, violent death, and suicide rates by incarceration time.** Instantaneous death rate among incarcerated men and boys in for all causes of death (black), violent deaths (purple), or suicide (yellow) in **(A)** closed prisons, **(B)** semi-open prisons, **(C)** police stations, and **(D)** youth detention. Bands indicate the 95% CI. Y-axis is shown in log10 scale and does not start at 1. Suicide rate not shown for semi-open prisons or youth detention due to the low number of suicides in these facilities. CI, confidence interval.
(PDF)

**S9 Fig. Age-standardized rates by incarceration time.** All-cause and cause-specific mortality rates per 100,000 person-years during and after the first 1 to 3 months of incarceration in each facility type. Rates were directly standardized to the age structure in the later period. Error bars indicate the 95% CI. IRRs between the earlier and later period were computed; significant differences are indicated (*$p < 0.05$; **$p < 0.01$). Causes for which there were fewer than 4 total deaths per time period were combined with other/unknown causes. 100k, 100,000 person-years; CI, confidence interval; comm diseases, communicable diseases; IRR, incidence rate ratio; non-comm diseases, noncommunicable diseases.
(PDF)

**S10 Fig. All-cause and cause-specific mortality rates among women following release from incarceration. (A)** Age-specific mortality rates for formerly incarcerated women. IRRs were computed relative to non-incarcerated female Mato Grosso do Sul residents. IRR, incidence rate ratio; *N*, number of deaths. **(B)** Instantaneous crude death rate among formerly incarcerated women for all causes of death (black), violence and suicide (purple), or causes other than violence and suicide (yellow). Bands indicate the 95% CI. Dotted horizontal line indicates the age-standardized mortality rate among non-incarcerated female residents, standardized to the age structure of the female formerly incarcerated population. 100K, 100,000 person-years; CI,

confidence interval; IRR, incidence rate ratio; *N*, number of deaths.
(PDF)

**S11 Fig. Association between cumulative time incarcerated and violent death post-release.**
**(A)** Kaplan–Meier estimates of survival from violence or other causes for men post-release,
stratified by total time incarcerated in any facility type. *p*-Values correspond to the log-rank
test. **(B)** Stacked bar plots showing the association between total time incarcerated and propor-
tions of violent deaths among men post-release, stratified by age group. **(C)** Boxplots depicting
the association between total time incarcerated and post-release time to death (in years) for
violent deaths and other causes, stratified by age group. All analyses in this figure were per-
formed on a reduced cohort of men with consistent follow-up time (all censored at 5 years
post-release).
(PDF)

**S12 Fig. Sensitivity of mortality rate ratios to the inclusion of uncertain matches.** IRRs esti-
mates for age-standardized and age-specific, all-cause and cause-specific mortality among **(A)**
men and **(B)** women upon the exclusion (red) or inclusion (blue) of 345 matches for which
mother's name was missing and there were multiple perfect matches by name. Rate ratios are
depicted on a log10 scale. 100k, 100,000 person-years; comm diseases, communicable diseases;
IRR, incidence rate ratio; non-comm diseases, noncommunicable diseases.
(PDF)

**S13 Fig. Crude rates and proportions of cause-specific deaths in custody vary by state.** Bar
plots depicting the crude mortality rate and proportions of cause-specific deaths during incar-
ceration in each Brazilian state between 2016 and 2018, as reported by the Brazilian National
Prison Department (DEPEN). The vertical dotted line indicates the mean crude mortality rate
across all states. DEPEN, Departamento Penitenciário Nacional.
(PDF)

## Acknowledgments

We thank Agência Estadual de Administração do Sistema Penitenciário (AGEPEN) and the
Brazilian Ministry of Health for providing access to databases. We honor the lives of those
directly and indirectly impacted by incarceration in Mato Grosso do Sul.

## Author Contributions

**Conceptualization:** Yiran E. Liu, Julio Croda, Katharine S. Walter, Jason R. Andrews.

**Data curation:** Yiran E. Liu, Everton Ferreira Lemos, Katharine S. Walter.

**Formal analysis:** Yiran E. Liu.

**Funding acquisition:** Jason R. Andrews.

**Investigation:** Yiran E. Liu, Everton Ferreira Lemos, Crhistinne Cavalheiro Maymone Gonçal-
   ves, Roberto Dias de Oliveira, Andrea da Silva Santos, Agne Oliveira do Prado Morais, Mar-
   iana Garcia Croda, Maria de Lourdes Delgado Alves, Julio Croda, Katharine S. Walter,
   Jason R. Andrews.

**Methodology:** Yiran E. Liu, Katharine S. Walter, Jason R. Andrews.

**Project administration:** Jason R. Andrews.

**Resources:** Jason R. Andrews.

**Software:** Yiran E. Liu.

**Supervision:** Julio Croda, Katharine S. Walter, Jason R. Andrews.

**Validation:** Yiran E. Liu, Everton Ferreira Lemos, Katharine S. Walter.

**Visualization:** Yiran E. Liu.

**Writing – original draft:** Yiran E. Liu, Katharine S. Walter, Jason R. Andrews.

**Writing – review & editing:** Yiran E. Liu, Everton Ferreira Lemos, Crhistinne Cavalheiro
Maymone Gonçalves, Roberto Dias de Oliveira, Andrea da Silva Santos, Agne Oliveira do
Prado Morais, Mariana Garcia Croda, Maria de Lourdes Delgado Alves, Julio Croda,
Katharine S. Walter, Jason R. Andrews.

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
