## [Editor Report · Decision Letter 0]

24 May 2021

Dear Dr Andrews, 

Thank you for submitting your manuscript entitled "All-cause and cause-specific mortality during and following incarceration in Brazil" for consideration by PLOS Medicine.

Your manuscript has now been evaluated by the PLOS Medicine editorial staff as well as by an academic editor with relevant expertise and I am writing to let you know that we would like to send your submission out for external peer review.

Please re-submit your manuscript within two working days, i.e. by May 26 2021 11:59PM.

Kind regards,

Callam Davidson

Associate Editor

PLOS Medicine

---

## [Decision Letter · Decision Letter 1]

9 Jul 2021

Dear Dr. Andrews,

Thank you very much for submitting your manuscript "All-cause and cause-specific mortality during and following incarceration in Brazil" (PMEDICINE-D-21-02293R1) for consideration at PLOS Medicine. 

[LINK]

In light of these reviews, we will not be able to accept the manuscript for publication in the journal in its current form, but we would like to consider a revised version that addresses the reviewers' and editors' comments. You will understand that we cannot make any decision about publication until we have seen the revised manuscript and your response, and we plan to seek re-review by one or more of the reviewers. 

We hope to receive your revised manuscript by Jul 30 2021 11:59PM. Please email us (plosmedicine@plos.org) if you have any questions or concerns.

We look forward to receiving your revised manuscript and please get in touch with any questions. 

Sincerely,

Callam Davidson, 

Associate Editor

PLOS Medicine

plosmedicine.org

Title: Please update the title to match PLOS Medicine’s typical format (see https://journals.plos.org/plosmedicine/ for examples). ‘All-cause and cause-specific mortality during and following incarceration in Brazil: a registry linkage and modelling study’ or similar, for example.

Abstract: Please add a final line to the ‘Methods and Findings’ section of the abstract that begins “The limitations of this study are” and summarises the main limitations. Please also ensure any conclusions are written in the past tense. 

References: Please format these to remove and bold or italicised text and list the first six authors before ‘et al.’.

Please remove the ‘Author contributions’ section from the end of the manuscript as, in the event of publication, this information is presented in the metadata based on your responses to the submission form.

If a protocol or statistical analysis plan were prepared in advance of the study, please can these be provided as supplementary materials and referenced appropriately in the methods (if an analysis plan was prepared, please highlight any analyses that were not pre-specified).

Please include an ‘Author Summary’ section as outlined here https://journals.plos.org/plosmedicine/s/revising-your-manuscript

Please consider whether the term ‘sex’ would be more appropriate than ‘gender’ throughout the manuscript. 

Please include line numbering in the margin. 

Comments from the reviewers:

Reviewer #1: This is a useful and well-conducted study on all-cause and cause-specific mortality during and following incarceration in Brazil. The study design, datasets, and statistical methods and analyses are mostly adequate. However, there are still a few issues needing attention.

1) It said all rates were age-standardised, but why not both age and sex standardised as there were female prisoners too?

2) In statistical analysis section, it says "Deaths of individuals under 18 years of age were excluded from rates analysis due to missing age structure information for those under 18". As the main analyses were on rates, does this mean the study cohort should be defined as those over18 years old? It's a bit confusing throughout the paper, in text, tables and figures that some are on 18+, some on 15+ and some on 16+? 

3) There is no table at all in the main text. Need a comprehensive and informative table 1 on summary of the dataset on baselines (age, sex, and etc) and possibly outcomes of interests.

4) Regarding figures, Figure 2 can be moved to supplementary as mainly details on data linkage. Figure 4 presents the main results of the paper but why there was no "closed prison (women)"? 

5) The deaths trend analysis after release as shown in Figure 5 is very interesting but lacks of solid interpretation on the reasons behind the trend which should be supported by the literature and statistics of trends in risk factors.

6) In conclusions in the abstract, it says "The cause-specific trends and risk of incarceration-associated mortality in Brazil differ from those reported in high-income countries...". However, this is not a comparative study, and can't compare like for like, therefore it's not adequate to conclude this as a key conclusion of the study.

Reviewer #2: "All-cause and cause-specific mortality during and following incarceration in Brazil." 

This is an interesting and relevant study examining mortality rates among criminal justice-involved populations in the Brazilian state, Mato Grosso do Sul. The authors find elevated mortality rates among this population, with particularly alarming rates of death due to violence both while under supervision and post-release. 

1. My main issues with the manuscript as it stands lie in the framing of the study and the discussion of findings. 

a. With regard to the former, the paper is framed as a study of "incarceration-associated mortality" (see abstract and introduction). However, this is a misnomer in a couple of ways. For one, the paper does not and cannot trace deaths to being caused by or related to incarceration itself. These are death rates among justice-involved persons- NOT incarceration-associated mortality. Second, the study also includes mortality rates of those in police stations. Is this being defined as incarceration as well? Please clarify. 

b. As for the latter, and this is a similar issue to the one abovementioned, the discussion makes problematic statements, but most notably: 

"These disparities [in mortality] may be partially explained by pre-existing vulnerabilities in system-impacted populations, but likely also reflect the long-term negative effects of incarceration due to communicable diseases, chronic stress, social deprivation, malnutrition, and/or medical neglect, as well as the social conditions associated with previous incarceration, such as stigma, poverty, and barriers to housing, healthcare and employment." 

 First, the author cannot establish causality between exposure to carceral settings and death, or causes of death. Second, the study does not speaks to this swath of potential hypotheses and variables. Third, things like communicable diseases were actually pretty comparable across post-release and general populations in the findings. 

Importantly, the authors should acknowledge that they cannot establish causality, let alone speculate about underlying reasons for causality. Also, justice-involved populations exhibit riskier lifestyles, including violent behavior, gang involvement, and substance use - all of which may contribute to these disparities in mortality. 

2. The causes of death while incarcerated piqued my curiosity in terms of the validity and reliability of these data. It strikes me that prison administrators and criminal justice organizations might have incentive to manipulate the data in this case - for example, claiming a death was a suicide rather than due to violence perpetrated by another prisoner or by a guard. Can the authors speak a little bit more about whether there might be reliability or validity issues of this nature? 

3. I would appreciate more discussion of the differences between data culled from DEPEN vs. SIGO. By that I do not mean how well records were matched, but more so - do these agencies have different missions, responsibilities, how do they collect these data similarly/differently? 

4. It strikes me that this research runs counter to some work on United States prison populations, which finds that being in prison can be protective for some populations in terms of mortality. This might be worth a discussion. For example, please see the work of Chris Wildeman and Evelyn Patterson. 

5. In general, the findings section was hard to follow and seemed unstructured/disorganized. I recommend cutting back on the amount of acronyms used and to organize the discussion of findings in order of interesting or meaningful comparisons. For example, a paragraph about gender differences, a paragraph about differences across recently vs. not recently incarcerated persons, a paragraph about differences across facility types, etc.

6. The authors should not claim that their findings from one state in Brazil could be generalizable (see conclusion). This is simply incorrect from a methodological standpoint. Also, bolstering this claim by saying that the mortality rates in prison are similar to other places in Brazil is a misleading justification of this point. 

Reviewer #3: The following paper is ambitious in its goal to provide overall and cause-specific estimates of death rates during and following incarceration in Brazil. These estimates were compared to non-incarcerated residents Nevertheless, by linking the national mortality database and prison records, the authors found that deaths in custody were more than twice as high as deaths reported. While I commend the authors for breadth of the work, I found myself confused at times, regarding which populations were included or excluded from the analyses. Further, the methodology utilized to undertake the analysis requires much more detail than provided in the manuscript. To this end, citations would be helpful. My general recommendation is that the authors focus on mortality during incarceration (or after incarceration), but not both in the same paper. I found it overwhelming, and limiting it to such would allow the authors to dedicate more space to providing details regarding the methodology and underlying assumptions. Regardless, the findings add to this literature and I did think the results of the mortality levels during and after incarceration increase the knowledge to the general area. I enumerate my specific concerns regarding the manuscript below.

1. Describe how you projected the formerly incarcerated population in more detail. Right now, I understand that the authors stratified person-years by release year of the cohort, but I do not know anything about how or why a person exited the formerly incarcerated population.

2. Relatedly, it is not clear how the authors distinguished formerly incarcerated individuals with recent incarceration from the comparison group of the general population.

3. Please explain more details regarding the estimation of the age structure. 

4. For the formerly incarcerated persons, the population only seems to change over time when persons die. However, there is certainly the risk of reincarceration. You mention this as means for individuals to be censored in the Pre-processing of SIGO incarceration database section, but it is not clear. I am guessing that the prison data contained this information at the individual level, however, my understanding is that you used summary level data. Please help me understand.

5. The differences in causes of death by facility make me question the assumption regarding each facility having the same age structure. Is there another option? That is, do you have wide age bins for each facility type?

6. I noted above that at times I got lost regarding the population included or excluded in analyses. I think this can easily be solved if the authors reiterate the population included/excluded. It would also be helpful, if the authors decide to keep both analyses, to partition the details of each piece in the data & methods and results sections. For example, describe the data and methods for the estimation of rates during incarceration, and then separately describe the data and methods for the post-incarceration mortality. It may be that the majority of my concerns can be taken care of by doing this.

7. I keep coming back to the decision to censor individuals upon movement to a different facility. It makes sense when thinking about facility specific death rates. However, part of the argument regarding deaths relates to the exposure to the settings, which makes me wonder about cumulative exposure in correctional settings. Further, one of the issues above regarding the formerly incarcerated population becomes less clear. For each population, does the movement censor the observation? It cannot because there are the release cohorts. I think I just need clearer depiction in the text.

8. In the abstract, the authors use the term "excess incarceration-associated mortality". When I read the term excess with mortality, I immediately think of deaths in excess of the expected. I do not think the authors seek to communicate such primarily because they do not do such, my understanding is that their study sets the foundation for future work. That is, this study serves as a baseline. I understand that the death rates found exceed that reported by the government, but as the authors also state, this is due to underreporting and insufficient data.

[LINK]

---

## [Decision Letter · Decision Letter 2]

23 Aug 2021

Dear Dr. Andrews,

Thank you very much for re-submitting your manuscript "All-cause and cause-specific mortality during and following incarceration in Brazil: a retrospective cohort study" (PMEDICINE-D-21-02293R2) for review by PLOS Medicine.

I have discussed the paper with my colleagues and the academic editor and it was also seen again by the reviewers. I am pleased to say that provided the remaining editorial and production issues are dealt with we are planning to accept the paper for publication in the journal.

[LINK]

Please also check the guidelines for revised papers at http://journals.plos.org/plosmedicine/s/revising-your-manuscript for any that apply to your paper. 

We look forward to receiving the revised manuscript by Aug 30 2021 11:59PM.   

Sincerely,

Callam Davidson, 

Associate Editor 

PLOS Medicine

plosmedicine.org

Requests from Editors:

Please state that your study did not have a pre-specified analysis plan in the methods section.

Please include your study design (retrospective cohort study) in both your abstract and methods, as it currently only appears in your title and your conclusions.

Line 45: The final line of the background section of your abstract should clearly state the study question. Please add an additional line to this effect. 

Line 114: Please remove the second bullet point in the ‘What do these findings mean?’ section of your author summary, as I feel it is too broad a statement to be included here and is better placed in the discussion where a more nuanced approach can be taken.

Line 566: Please include ‘to our knowledge’ whenever making claims of primacy (‘As the first systematic study’). 

Figure S9: Please define what your error bars show in the figure legend. 

References 20, 28, and 35: Insufficient material is presented to allow the reader to locate these sources – please provide more detail and cite using Vancouver style (see guidelines here: https://journals.plos.org/plosmedicine/s/submission-guidelines#loc-references) 

Reference 52: Please cite preprints per the guidelines here -https://journals.plos.org/plosmedicine/s/submission-guidelines#loc-references

Please ensure that the study is reported according to the STROBE guideline, and include the completed STROBE checklist as Supporting Information. Please add the following statement, or similar, to the Methods: "This study is reported as per the Strengthening the Reporting of Observational Studies in Epidemiology (STROBE) guideline (S1 Checklist)."

Comments from Reviewers:

Reviewer #1: Many thanks authors for their great effort to improve the manuscript. The authors have addressed my comments professionally. I am satisfied with the response and revision. Only a minor point, for table 1. all the categorical variables need to be summarised as count and percentage e.g. xx (yy%). Please just correct this and no need to come back to me. No other issues needing attention and I recommend publicaiton of the paper.

Reviewer #2: I appreciate the authors' attention to my comments, especially my concerns about causality. I am satisfied with the revisions they have carried out.

[LINK]

---

## [Editor Report · Decision Letter 3]

1 Sep 2021

Dear Dr Andrews, 

On behalf of my colleagues and the Academic Editor, Dr Vikram Patel, I am pleased to inform you that we have agreed to publish your manuscript "All-cause and cause-specific mortality during and following incarceration in Brazil: a retrospective cohort study" (PMEDICINE-D-21-02293R3) in PLOS Medicine.

PRESS

Sincerely, 

Callam Davidson 

Associate Editor 

PLOS Medicine